# MULTI-OBJECTIVE MULTI-AGENT REINFORCEMENT LEARNING WITH PARETO-STATIONARY CONVERGENCE

## ABSTRACT

Multi-objective multi-agent reinforcement learning (MOMARL) problems frequently arise in real world applications (e.g., path planning for robots) but have not been explored well. To find Pareto-optimum is NP-hard, and thus some multi-objective algorithms have emerged recently to provide Pareto-stationary solution centrally, managed by a single agent. Yet, they cannot deal with MOMARL problem, as the dimension of global state-action $(\boldsymbol{s}, \boldsymbol{a})$ grows exponentially with the number of spatially distributed agents. To tackle this issue, we design a novel graph-truncated $Q$-function approximation method for each agent $i$, which does not require the global state-action $(\boldsymbol{s}, \boldsymbol{a})$ but only the neighborhood state-action $(s_{\mathcal{N}_i^\kappa}, a_{\mathcal{N}_i^\kappa})$ of its $\kappa$-hop neighbors. To further reduce the dimension to state-action $(s_{\mathcal{N}_i^\kappa}, a_i)$ with only local action, we further develop a concept of action-averaged $Q$-function and establish the equivalence between using graph-truncated $Q$-function and action-averaged $Q$-function for policy gradient approximation. Accordingly, we develop a distributed scalable algorithm with linear function approximation and prove that it successfully converges Pareto-stationary solution at rate $\mathcal{O}(1/T)$ that is inversely proportional to time domain $T$. Finally, we run simulations in a robot path planning environment and show our algorithm converges to greater multi-objective values as compared to the latest MORL algorithm, and performs close to the central optimum with much shorter running time.

## 1 INTRODUCTION

As real-world applications become increasingly complex, multi-objective optimization problems are becoming more prevalent. For example, in the e-commerce domain (Weck et al., 2022; Xu et al., 2024), platforms aim for product recommendations that are not only clickable and purchasable but also engaging enough to encourage user sharing and collection. This scenario involves optimizing multiple objectives, including the click-through rate, purchase rate, and collection rate of the products. For such scenarios involving multiple optimization objectives, the traditional setting of a single reward structure in the reinforcement learning (RL) framework (Sutton & Barto, 1998) is obviously insufficient to describe. Therefore, it is necessary to establish multi-objective RL (MORL) problems.

Different from the rapid development of traditional RL (Grondman et al, 2012; Zhang et al, 2021), the research in MORL (Ge et al., 2022; Stamenkovic et al, 2022) is still in its infancy to address the potential conflicts between multiple objectives. One common approach to solving MORL problem involves assigning weights to different objectives and transforming the multi-objective problem into a single-objective problem (Blondin & Hale, 2020). However, this approach has the limitation of assuming known objective weights, which can restrict its applicability. In the MORL problems, a more appropriate and relevant metric is to find a Pareto-optimal solution for all objectives, where no objective can be unilaterally improved without sacrificing another. As many real-world MORL problems are typically non-convex, finding the Pareto-optimal solution is NP-hard (Yang et al., 2024).

To address the NP-hard nature of non-convex MORL problems, Pareto-stationary solutions (a necessary condition for Pareto optimality) are employed (Sener & Koltun, 2018). For the MORL problems with continuous action space, (Chen et al., 2021) proposed an actor-critic MORL algorithm based on the deterministic policy-gradient (Silver et al., 2014). More generally, for the MORL problem with non-continuous action space, a unified multi-objective actor-critic algorithmic framework was

proposed for both discounted and average reward settings in (Zhou et al., 2024), where the update of stochastic policy parameters employs the multi-gradient descent method in (Désidéri, 2012).

The aforementioned methods are all directed towards addressing the MORL problem in a centralized setting or for a single agent. However, practical applications of MORL problems often involve multi-agents. For instance, teams of robots need to decide themselves how to explore distinct regions by simultaneously minimizing energy consumption and travel time. In comparison to the MORL problem with single-agent, the multi-objective multi-agent problem (MOMARL) is more intricate as it encompasses not only potential conflicts among different objectives but also interactions between the distributed agents with limited communication. An intuitive approach to the MOMARL problem is to consider it as a MORL problem with a single agent, where the state and action are represented by the joint states and joint actions of all agents, respectively. However, as the number of agents increases, the size of their joint state-action space will exponentially grow. This characteristic renders the current algorithms used for solving MORL problems with a single agent in (Chen et al., 2021; Zhou et al., 2024) unsuitable for large-scale scenarios with multi-agents. Consequently, the MOMARL problem poses new challenges to the design of scalable algorithms and their theoretical analysis.

This paper aims to address the following problem: *How to develop a scalable algorithm for the MOMARL problem and ensure its convergence to Pareto-stationary of the multi-objective function?* The contributions of this paper are described as follows.

(i) In order to improve the scalability of the algorithm and avoid using the global state-action, we design a novel graph-truncated $Q$-function approximation for each agent $i$, which only requires the neighborhood state-action $(s_{\mathcal{N}_i^\kappa}, a_{\mathcal{N}_i^\kappa})$ of its $\kappa$-hop neighbors, instead of the global state-action. Additionally, we introduce a new concept of action-averaged $Q$-function and establish the equivalence between using the graph-truncated $Q$-function and action-averaged $Q$-function for policy gradient approximation.

(ii) Based on the concept of action-averaged $Q$-function, we propose a distributed scalable actor-critic algorithm for the MOMARL problem. In critic step, we use linear function to approximate the action-averaged $Q$-function, which further reduces the dimension of state-action to $(s_{\mathcal{N}_i^\kappa}, a_i)$ with local action. In addition, we use the multi-gradient descent method in actor step to update the policy parameter for finding a Pareto-stationary solution.

(iii) We prove that the proposed scalable algorithm for MOMARL successfully converges to the Pareto-stationary solution at rate $\mathcal{O}(1/T)$ that is inversely proportional to time domain $T$. Moreover, we run simulations in a robot path planning environment and show our algorithm converges to greater multi-objective values as compared to the latest MORL algorithm (Zhou et al., 2024), and performs close to the central optimum with much shorter running time.

## 2 THE NEW MOMARL PROBLEM FORMULATION AND PRELIMINARIES

### 2.1 MODEL OF THE MOMARL PROBLEM

The MOMARL problem can be described as $(\mathcal{N}, \mathcal{M}, \mathcal{G}(\mathcal{N}, \mathcal{E}), \{\mathcal{S}_i\}_{i\in\mathcal{N}}, \{\mathcal{A}_i\}_{i\in\mathcal{N}}, \{\mathcal{P}_i\}_{i\in\mathcal{N}}, \boldsymbol{\rho},$ $\{r_i^m\}_{i\in\mathcal{N},m\in\mathcal{M}}, \boldsymbol{\gamma})$, where $\mathcal{N} = \{1, \cdots, N\}$ and $\mathcal{M} = \{1, \cdots, M\}$ represent the agent set and the objective set, respectively. $\mathcal{G} = (\mathcal{N}, \mathcal{E})$ represents the communication network among agents with $\mathcal{E}$ being the set of edges [1]. For integer $\kappa \geq 1$, denote $\mathcal{N}_i^\kappa$ as the $\kappa$-hop neighborhood of agent $i$.

**State and action**: $\mathcal{S}_i$ and $\mathcal{A}_i$ represent the local state space and the local action space of agent $i$, respectively. Denote $\boldsymbol{\mathcal{S}} = \prod_{i=1}^N \mathcal{S}_i$ and $\boldsymbol{\mathcal{A}} = \prod_{i=1}^N \mathcal{A}_i$ as the global state space and the global action space, respectively. Denote $\boldsymbol{s} = (s_1, \cdots, s_N) \in \boldsymbol{\mathcal{S}}$ and $\boldsymbol{a} = (a_1, \cdots, a_N) \in \boldsymbol{\mathcal{A}}$ as the global state and the global action of agents, where $s_i \in \mathcal{S}_i$ and $a_i \in \mathcal{A}_i$ represent the local state and local action of agent $i \in \mathcal{N}$, respectively. For integer $\kappa \geq 1$, denote $s_{\mathcal{N}_i^\kappa}$ and $a_{\mathcal{N}_i^\kappa}$ as the state and action of agent $i$'s $\kappa$-hop neighbors, respectively. Moreover, denote $\mathcal{S}_{\mathcal{N}_i^\kappa} = \prod_{j\in\mathcal{N}_i^\kappa} \mathcal{S}_j$ and $\mathcal{A}_{\mathcal{N}_i^\kappa} = \prod_{j\in\mathcal{N}_i^\kappa} \mathcal{A}_j$ as the state space and the action space of agent $i$'s $\kappa$-hop neighbors, respectively.

---

[1]For the case of time-varying neighbor agents, our algorithm is still applicable if the agent communicates intermittently (or delays communication) with its initial neighbor. In the process of convergence analysis of the algorithm, we just need to introduce an additional error term caused by communication disconnection or delay.

**State transition probability function**: $\mathcal{P}_i(s_i'|s_{\mathcal{N}_i^1}, a_i) : \mathcal{S}_{\mathcal{N}_i^1} \times \mathcal{A}_i \times \mathcal{S}_i \rightarrow [0, 1]$ is the state transition probability function of agent $i$, dependent of its 1-hop neighborhood state and its local action. Denote $\boldsymbol{\mathcal{P}}(s'|s, a) = \prod_{i=1}^{N} \mathcal{P}_i(s_i'|s_{\mathcal{N}_i^1}, a_i) : \boldsymbol{\mathcal{S}} \times \boldsymbol{\mathcal{A}} \times \boldsymbol{\mathcal{S}} \rightarrow [0, 1]$ as the global state transition probability function. Note that the definition of the state transition probability function $\prod_{i=1}^{N} \mathcal{P}_i(s_i'|s_{\mathcal{N}_i^1}, a_i)$ is common in the literature. For example, it applies to the scenario of traffic signal control problem (Chu et al., 2020; Dai et al., 2024), where the traffic flow at each intersection is influenced by the traffic flow at its neighboring intersections and its own signal light.

**Initial state distribution**: $\rho$ is the distribution of the initial state $s_0$.

**Reward function**: $r_i^m(s_i, a_i) : \mathcal{S}_i \times \mathcal{A}_i \rightarrow \mathbb{R}$ is the reward function of agent $i \in \mathcal{N}$ in the objective $m \in \mathcal{M}$. Denote $\boldsymbol{s}_t = (s_{1,t}, \cdots, s_{N,t})$ and $\boldsymbol{a}_t = (a_{1,t}, \cdots, a_{N,t})$ as the global state and the global action at time $t$, respectively. The reward of agent $i \in \mathcal{N}$ in the objective $m \in \mathcal{M}$ at time $t$ can be represented as $r_{i,t}^m = r_i^m(s_{i,t}, a_{i,t})$, as in the literature (Chu et al., 2020; Dai et al., 2024; Zhou et al., 2023; Qu et al., 2020a).

**Discount factor**: $\boldsymbol{\gamma} = (\gamma^1, \cdots, \gamma^M)^\top \in \mathbb{R}^M$ with $\gamma^m \in (0, 1)$ being the discount factor in the objective $m \in \mathcal{M}$.

**Softmax policy**: In this paper, we use the parameterized softmax policy $\pi_{\theta_i}(a_i|s_i)$ with parameter $\theta_i \in \mathbb{R}^{|\mathcal{S}_i||\mathcal{A}_i|}$, which is described as

$$\pi_{\theta_i}(a_i|s_i) = \frac{\exp(\theta_{i,s_i,a_i})}{\sum_{a_i'} \exp(\theta_{i,s_i,a_i'})}, \tag{1}$$

where $\theta_{i,s_i,a_i}$ represents the element corresponding to $(s_i, a_i)$ in $\theta_i$. Denote $\boldsymbol{\theta} = (\theta_1^\top, \cdots, \theta_N^\top)^\top \in \mathbb{R}^{\sum_{i=1}^{N} |\mathcal{S}_i||\mathcal{A}_i|}$ as the joint policy parameter of agents and $\boldsymbol{\pi_\theta}(a|s) = \prod_{i=1}^{N} \pi_{\theta_i}(a_i|s_i)$ be the joint policy of all agents. Note that the softmax policy is used in RL to ensure the exploration of agents (Zhou et al., 2023; Zhang et al., 2022).

In the MOMARL problem, given a joint policy parameter $\boldsymbol{\theta}$, the $m$-th objective of all agents is defined as $J^m(\boldsymbol{\theta})$ and represented as

$$J^m(\boldsymbol{\theta}) = \mathbb{E}_{\boldsymbol{s} \sim \boldsymbol{\rho}}\Big[\frac{1}{N} \sum_{t=0}^{\infty} \sum_{i=1}^{N} (\gamma^m)^t r_{i,t}^m | \boldsymbol{s}_0 = \boldsymbol{s}, \boldsymbol{a}_t \sim \boldsymbol{\pi_\theta}(\cdot|\boldsymbol{s}_t)\Big]. \tag{2}$$

The goal of agents in the MOMARL problem is to find a joint policy parameter $\boldsymbol{\theta}$ to maximize the following composite objective, i.e.,

$$\max_{\boldsymbol{\theta}} \boldsymbol{J}(\boldsymbol{\theta}) = [J^1(\boldsymbol{\theta}), \cdots, J^M(\boldsymbol{\theta})]^\top \in \mathbb{R}^M. \tag{3}$$

In order to address the potential conflicts among the $\boldsymbol{J}(\boldsymbol{\theta})$ in (3), the notions of Pareto-optimality and $\epsilon$-Pareto-stationarity are introduced as follows.

**Definition 1** *(Pareto-optimality) A solution $\boldsymbol{\theta}$ dominates solution $\boldsymbol{\theta}'$ if and only if $J^m(\boldsymbol{\theta}) \geq J^m(\boldsymbol{\theta}')$, $\forall m \in \mathcal{M}$ and $\exists m' \in \mathcal{M}$, $J^{m'}(\boldsymbol{\theta}) > J^{m'}(\boldsymbol{\theta}')$. A solution $\boldsymbol{\theta}$ is Pareto-optimal if it is not dominated by any other solution.*

Considering that finding Pareto-optimal solutions for non-convex MOMARL problems is NP-hard, it is generally more practical to seek the $\epsilon$-Pareto-stationary solution instead of the Pareto-optimal solution (Kumar et al., 2019).

**Definition 2** *($\epsilon$-Pareto-stationarity) A solution $\boldsymbol{\theta}$ is $\epsilon$-Pareto stationary if there exists $\boldsymbol{\lambda} = (\lambda^1, \cdots, \lambda^M)^\top \in \mathbb{R}^M$ such that $\min_{\boldsymbol{\lambda} \in \mathbb{R}^M} \|\nabla_{\boldsymbol{\theta}} \boldsymbol{J}(\boldsymbol{\theta})^\top \boldsymbol{\lambda}\|_2^2 \leq \epsilon$ with $\boldsymbol{\lambda} \geq 0$, $\|\boldsymbol{\lambda}\|_1 = 1$, and $\epsilon > 0$.*

Based on Definitions 1-2, it is obvious that the Pareto-stationarity is a necessary condition for a solution to be Pareto-optimal. Specifically, in the context of convex MOMARL problems, the solutions that are Pareto-stationary also qualify as Pareto-optimal. Given the complexity associated with the MOMARL problem, this paper focuses on developing a distributed scalable algorithm to identify and achieve Pareto-stationarity.

## 2.2 Preliminaries in the MOMARL problem

In the MOMARL problem, for any joint policy parameter $\boldsymbol{\theta}$ and $m \in \mathcal{M}$, the global $Q$-function $Q^m(\boldsymbol{s}, \boldsymbol{a}; \boldsymbol{\theta})$ in $m$-th objective is defined as

$$Q^m(\boldsymbol{s}, \boldsymbol{a}; \boldsymbol{\theta}) = \mathbb{E}_{\boldsymbol{\pi_\theta}}\Big[ \frac{1}{N} \sum_{t=0}^{\infty} \sum_{i=1}^{N} (\gamma^m)^t r_{i,t}^m | s_0 = \boldsymbol{s}, \boldsymbol{a}_0 = \boldsymbol{a} \Big]. \tag{4}$$

Different from the definition of the global $Q$-function in (4), for each agent $i \in \mathcal{N}$, its local $Q$-function $Q_i^m(\boldsymbol{s}, \boldsymbol{a}; \boldsymbol{\theta})$ in $m$-th objective is defined as

$$Q_i^m(\boldsymbol{s}, \boldsymbol{a}; \boldsymbol{\theta}) = \mathbb{E}_{\boldsymbol{\pi_\theta}}\Big[ \sum_{t=0}^{\infty} (\gamma^m)^t r_{i,t}^m | s_0 = \boldsymbol{s}, \boldsymbol{a}_0 = \boldsymbol{a} \Big]. \tag{5}$$

Based on the definitions of the global $Q$-function (4) and the local $Q$-function (5), we have

$$Q^m(\boldsymbol{s}, \boldsymbol{a}; \boldsymbol{\theta}) = \frac{1}{N} \sum_{i=1}^{N} Q_i^m(\boldsymbol{s}, \boldsymbol{a}; \boldsymbol{\theta}), \tag{6}$$

which shows the global $Q$-function can be decomposed into the sum of the local $Q$-functions of all agents. In the MOMARL problem, given the joint policy parameter $\boldsymbol{\theta}$, define $d_{\boldsymbol{\rho}}^{\boldsymbol{\theta},m}(\boldsymbol{s})$ as the discounted state visitation distribution, which is represented as

$$d_{\boldsymbol{\rho}}^{\boldsymbol{\theta},m}(\boldsymbol{s}) = (1 - \gamma^m) \sum_{t=0}^{\infty} (\gamma^m)^t \mathrm{Pr}^{\boldsymbol{\pi_\theta}}(\boldsymbol{s}_t = \boldsymbol{s} | \boldsymbol{s}_0 \sim \boldsymbol{\rho}), \tag{7}$$

where $\mathrm{Pr}^{\boldsymbol{\pi_\theta}}(\boldsymbol{s}_t = \boldsymbol{s} | \boldsymbol{s}_0 \sim \boldsymbol{\rho})$ represents the probability of $\boldsymbol{s}_t = \boldsymbol{s}$ at time $t$ under the initial state distribution $\boldsymbol{\rho}$ and the joint policy $\boldsymbol{\pi_\theta}$. Moreover, let $\xi_{\boldsymbol{\rho}}^{\boldsymbol{\theta},m}(\boldsymbol{s}, \boldsymbol{a})$ be the discounted state-action visitation distribution of $(\boldsymbol{s}, \boldsymbol{a}) \in \mathcal{S} \times \mathcal{A}$ and satisfy

$$\xi_{\boldsymbol{\rho}}^{\boldsymbol{\theta},m}(\boldsymbol{s}, \boldsymbol{a}) = d_{\boldsymbol{\rho}}^{\boldsymbol{\theta},m}(\boldsymbol{s})\boldsymbol{\pi_\theta}(\boldsymbol{a}|\boldsymbol{s}). \tag{8}$$

In the MOMARL problem, some assumptions are introduced in the following.

**Assumption 1** *In the MOMARL problem, for any joint policy parameter $\boldsymbol{\theta}$ and objective $m \in \mathcal{M}$, $\xi_{\boldsymbol{\rho}}^{\boldsymbol{\theta},m}(\boldsymbol{s}, \boldsymbol{a})$ satisfies that*

$$\inf_{\boldsymbol{\theta}} \min_{(\boldsymbol{s},\boldsymbol{a}) \in \mathcal{S} \times \mathcal{A}} \xi_{\boldsymbol{\rho}}^{\boldsymbol{\theta},m}(\boldsymbol{s}, \boldsymbol{a}) > 0. \tag{9}$$

**Assumption 2** *In the MOMARL problem, for any agent $i \in \mathcal{N}$ and objective $m \in \mathcal{M}$, there exists constant $R > 1$ such that the instantaneous reward $r_{i,t}^m$ at time $t \geq 0$ satisfies $|r_{i,t}^m| \leq R$.*

Assumption 1 ensures that for any joint policy $\boldsymbol{\pi_\theta}$, $(\boldsymbol{s}, \boldsymbol{a}) \in \mathcal{S} \times \mathcal{A}$ is visited with a non-zero probability and Assumption 2 provides an upper bound on the reward. These assumptions are standard prerequisite for the convergence analysis of RL algorithms and can be found in (Zhou et al., 2023; Zhang et al., 2022).

Recall that the policy gradient theorem (Sutton et al., 2000) is the foundation of algorithm design in RL. Inspired by the theorem, in our MOMARL problem, we also have the following policy gradient lemma.

**Lemma 1** *In the MOMARL problem, for any joint policy parameter $\boldsymbol{\theta}$, the gradient of $J^m(\boldsymbol{\theta})$ in $m$-the objective with respect to $\boldsymbol{\theta}$ is given by:*

$$\nabla_{\boldsymbol{\theta}} J^m(\boldsymbol{\theta}) = \frac{1}{1 - \gamma^m} \mathbb{E}_{\boldsymbol{s} \sim d_{\boldsymbol{\rho}}^{\boldsymbol{\theta},m}, \boldsymbol{a} \sim \boldsymbol{\pi_\theta}} [\nabla_{\boldsymbol{\theta}} \log \boldsymbol{\pi_\theta}(\boldsymbol{a}|\boldsymbol{s}) Q^m(\boldsymbol{s}, \boldsymbol{a}; \boldsymbol{\theta})], \forall m \in \mathcal{M}. \tag{10}$$

Lemma 1 shows that the calculation of the policy gradient $\nabla_{\boldsymbol{\theta}} J^m(\boldsymbol{\theta})$ depends on $Q^m(\boldsymbol{s}, \boldsymbol{a}; \boldsymbol{\theta})$, which involves global state-action $(\boldsymbol{s}, \boldsymbol{a})$. Consequently, there are two challenges in applying (10): (i) the computational complexity of handling the global state-action $(\boldsymbol{s}, \boldsymbol{a})$ in a centralized setting is high; (ii) it is difficult to achieve efficient distributed decision making among multi-agents with limited communication.

# 3 DISTRIBUTED SCALABLE ACTOR-CRITIC ALGORITHM FOR MOMARL PROBLEM

In order to mitigate the RL algorithm's dependence on global state-action $(s, a)$, this section designs a distributed scalable algorithm through the following 3 steps as in Fig. 1: (1) We first propose a new **graph-truncated** $Q$-**function** approximation for each agent $i \in \mathcal{N}$, which does not require the global state-action $(s, a)$ but only the neighborhood state-action $(s_{\mathcal{N}_i^\kappa}, a_{\mathcal{N}_i^\kappa})$ of its $\kappa$-hop neighbors; (2) Then, we introduce a new concept of **action-averaged** $Q$-**function** and establish the equivalence between using the graph-truncated $Q$-function and action-averaged $Q$-function for policy gradient approximation; (3) Finally, we use **linear function** to approximate the action-averaged $Q$-function and reduce the dimensionality of state-action of each agent $i \in \mathcal{N}$ to $(s_{\mathcal{N}_i^\kappa}, a_i)$.

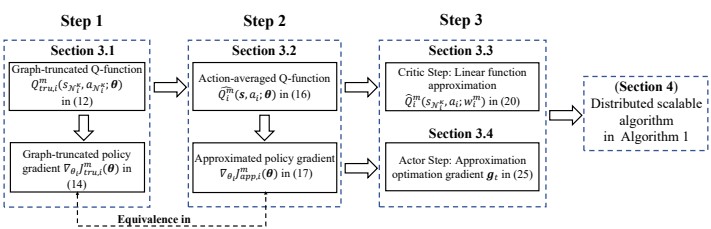

Figure 1: The main flowchart of algorithm design: Step 1 proposes a new graph-truncated $Q$-function $Q_{tru,i}^m(s_{\mathcal{N}_i^\kappa}, a_{\mathcal{N}_i^\kappa}; \boldsymbol{\theta})$ and the graph-truncated policy gradient $\nabla_{\theta_i} J_{tru,i}^m(\boldsymbol{\theta})$; Step 2 designs a action-averaged $Q$-function $\widehat{Q_i^m}(s, a_i; \boldsymbol{\theta})$ and approximation policy gradient $\nabla_{\theta_i} J_{app,i}^m(\boldsymbol{\theta})$, which is equivalent to $\nabla_{\theta_i} J_{tru,i}^m(\boldsymbol{\theta})$ (i.e., Proposition 1); Step 3 proposes the linear function approximation and policy parameter update for the distributed scalable algorithm in Section 4.

## 3.1 GRAPH-TRUNCATED $Q$-FUNCTION

In the following, we first introduce the formal definition of the exponential decay property in the MOMARL problem.

**Definition 3** *The MOMARL satisfies the $(\boldsymbol{\vartheta}, \boldsymbol{\varrho})$-exponential decay property with $\boldsymbol{\vartheta} = (\vartheta^1, \cdots, \vartheta^M)^\top \in \mathbb{R}^M, \boldsymbol{\varrho} = (\varrho^1, \cdots, \varrho^M)^\top \in \mathbb{R}^M$, if for any joint policy $\boldsymbol{\pi_\theta}$, agent $i \in \mathcal{N}$, objective $m \in \mathcal{M}$, $s_{\mathcal{N}_i^\kappa} \in \mathcal{S}_{\mathcal{N}_i^\kappa}$, $a_{\mathcal{N}_i^\kappa} \in \mathcal{A}_{\mathcal{N}_i^\kappa}$, $s_{-\mathcal{N}_i^\kappa}, s'_{-\mathcal{N}_i^\kappa} \in \mathcal{S}_{-\mathcal{N}_i^\kappa}$, and $a_{-\mathcal{N}_i^\kappa}, a'_{-\mathcal{N}_i^\kappa} \in \mathcal{A}_{-\mathcal{N}_i^\kappa}$, $Q_i^m(s, a; \boldsymbol{\theta})$ satisfies*

$$\left| Q_i^m(s_{\mathcal{N}_i^\kappa}, s_{-\mathcal{N}_i^\kappa}, a_{\mathcal{N}_i^\kappa}, a_{-\mathcal{N}_i^\kappa}; \boldsymbol{\theta}) - Q_i^m(s_{\mathcal{N}_i^\kappa}, s'_{-\mathcal{N}_i^\kappa}, a_{\mathcal{N}_i^\kappa}, a'_{-\mathcal{N}_i^\kappa}; \boldsymbol{\theta}) \right| \le \vartheta^m (\varrho^m)^{\kappa+1}. \quad (11)$$

The exponential decay property of the MOMARL problem indicates that the dependence of agent $i$'s local $Q$-function $Q_i^m(s, a; \boldsymbol{\theta})$ on other agents shrinks rapidly as the distance between them increases. By Assumption 2, we can directly obtain the following lemma.

**Lemma 2** *The MOMARL problem satisfies $\left( (\frac{R}{1-\gamma^1}, \cdots, \frac{R}{1-\gamma^M})^\top, \boldsymbol{\gamma} \right)$-exponential decay property.*

The proof can be found in Appendix A.1. Lemma 2 provides a possibility for agents to approximate $Q_i^m(s, a; \boldsymbol{\theta})$ by only using its $\kappa$-hop neighbors' information. Inspired by exponential decay property in Lemma 2, we design a proper class of graph-truncated $Q$-functions:

$$Q_{tru,i}^m(s_{\mathcal{N}_i^\kappa}, a_{\mathcal{N}_i^\kappa}; \boldsymbol{\theta}) = \sum_{s_{-\mathcal{N}_i^\kappa}, a_{-\mathcal{N}_i^\kappa}} \xi_{\boldsymbol{\rho}}^{\boldsymbol{\theta}, m}(s_{-\mathcal{N}_i^\kappa}, a_{-\mathcal{N}_i^\kappa} | s_{\mathcal{N}_i^\kappa}, a_{\mathcal{N}_i^\kappa}) Q_i^m(s_{\mathcal{N}_i^\kappa}, s_{-\mathcal{N}_i^\kappa}, a_{\mathcal{N}_i^\kappa}, a_{-\mathcal{N}_i^\kappa}; \boldsymbol{\theta}),$$

$$(12)$$

where $\xi_{\boldsymbol{\rho}}^{\boldsymbol{\theta}, m}(s_{-\mathcal{N}_i^\kappa}, a_{-\mathcal{N}_i^\kappa} | s_{\mathcal{N}_i^\kappa}, a_{\mathcal{N}_i^\kappa})$ is the weight coefficient and satisfies

$$\xi_{\boldsymbol{\rho}}^{\boldsymbol{\theta}, m}(s_{-\mathcal{N}_i^\kappa}, a_{-\mathcal{N}_i^\kappa} | s_{\mathcal{N}_i^\kappa}, a_{\mathcal{N}_i^\kappa}) = \frac{\xi_{\boldsymbol{\rho}}^{\boldsymbol{\theta}, m}(s_{\mathcal{N}_i^\kappa}, s_{-\mathcal{N}_i^\kappa}, a_{\mathcal{N}_i^\kappa}, a_{-\mathcal{N}_i^\kappa})}{\sum_{s'_{-\mathcal{N}_i^\kappa}, a'_{-\mathcal{N}_i^\kappa}} \xi_{\boldsymbol{\rho}}^{\boldsymbol{\theta}, m}(s_{\mathcal{N}_i^\kappa}, s'_{-\mathcal{N}_i^\kappa}, a_{\mathcal{N}_i^\kappa}, a'_{-\mathcal{N}_i^\kappa})}. \quad (13)$$

Using (12), we define the graph-truncated policy gradient $\nabla_{\theta_i} J_{tru,i}^m(\boldsymbol{\theta})$ as

$$\nabla_{\theta_i} J_{tru,i}^m(\boldsymbol{\theta}) = \frac{1}{1-\gamma} \mathbb{E}_{\boldsymbol{s} \sim d_{\boldsymbol{\rho}}^{\boldsymbol{\theta},m}, \boldsymbol{a} \sim \boldsymbol{\pi_\theta}} \Big[ \frac{1}{N} \sum_{j \in \mathcal{N}_i^\kappa} Q_{tru,j}^m(s_{\mathcal{N}_j^\kappa}, a_{\mathcal{N}_j^\kappa}; \boldsymbol{\theta}) \nabla_{\theta_i} \log \pi_{\theta_i}(a_i|s_i) \Big]. \quad (14)$$

The graph-truncated policy gradient approximation error is presented in the following.

**Lemma 3** *In the MOMARL problem, for any agent $i \in \mathcal{N}$ and objective $m \in \mathcal{M}$, we have*

$$\Big\| \nabla_{\theta_i} J_{tru,i}^m(\boldsymbol{\theta}) - \nabla_{\theta_i} J^m(\boldsymbol{\theta}) \Big\|_2 \leq \frac{\sqrt{2}R}{(1-\gamma^m)^2} (\gamma^m)^{\kappa+1}. \quad (15)$$

Similar to (Qu et al., 2020a), Lemma 3 shows that the graph-truncated $Q$-functions $\{Q_{tru,j}^m(s_{\mathcal{N}_j^\kappa}, a_{\mathcal{N}_j^\kappa}; \boldsymbol{\theta})\}_{j \in \mathcal{N}_i^\kappa}$ can effectively approximate the policy gradient $\nabla_{\theta_i} J^m(\boldsymbol{\theta})$ through the state-action $(s_{\mathcal{N}_i^\kappa}, a_{\mathcal{N}_i^\kappa})$. In order to improve the scalability of the algorithm, we further explore the properties of graph-truncated $Q$-function in (13) and reduce the dimensionality of the algorithm to $(s_{\mathcal{N}_i^\kappa}, a_i)$.

## 3.2 POLICY GRADIENT APPROXIMATION

To further reduce the neighbors' action $a_{\mathcal{N}_i^\kappa}$ in graph-truncated $Q$-function (12) to local action $a_i$, for any agent $i$ and objective $m$, we design a novel concept of "action-averaged $Q$-function" by using its $\kappa$-hop neighbors' rewards as follows:

$$\widehat{Q_i^m}(\boldsymbol{s}, a_i; \boldsymbol{\theta}) = \mathbb{E}_{\boldsymbol{\pi_\theta}} \Big[ \frac{1}{N} \sum_{t=0}^{\infty} (\gamma^m)^t \sum_{j \in \mathcal{N}_i^\kappa} r_j^m(s_{j,t}, a_{j,t}) | \boldsymbol{s}_0 = \boldsymbol{s}, a_{i,0} = a_i \Big]. \quad (16)$$

Define $\nabla_{\theta_i} J_{app}^m(\boldsymbol{\theta})$ as the approximated policy gradient of agent $i$ by using the action-averaged $Q$-function in (16), given by:

$$\nabla_{\theta_i} J_{app,i}^m(\boldsymbol{\theta}) = \frac{1}{1-\gamma^m} \mathbb{E}_{\boldsymbol{s} \sim d_{\boldsymbol{\rho}}^{\boldsymbol{\theta},m}, a_i \sim \pi_{\theta_i}} \Big[ \widehat{Q_i^m}(\boldsymbol{s}, a_i; \boldsymbol{\theta}) \nabla_{\theta_i} \log \pi_{\theta_i}(a_i|s_i) \Big]. \quad (17)$$

Unlike the graph-truncated policy gradient $\nabla_{\theta_i} J_{tru,i}^m(\boldsymbol{\theta})$ in (14) that requires $a_{\mathcal{N}_i^\kappa}$, (17) only requires the local action $a_i$. As shown in Fig. 1, we establish the equivalence between graph-truncated policy gradient $\nabla_{\theta_i} J_{tru,i}^m(\boldsymbol{\theta})$ and approximated policy gradient $\nabla_{\theta_i} J_{app}^m(\boldsymbol{\theta})$ in the following proposition.

**Proposition 1** *In the MOMARL problem, given a joint policy $\boldsymbol{\pi_\theta}$, for any agent $i \in \mathcal{N}$ and objective $m \in \mathcal{M}$, it holds*

$$\nabla_{\theta_i} J_{tru,i}^m(\boldsymbol{\theta}) = \nabla_{\theta_i} J_{app,i}^m(\boldsymbol{\theta}). \quad (18)$$

The proof of Proposition 1 can be found in Appendix A.3. Proposition 1 provides an equivalence between $Q_{tru,i}^m(s_{\mathcal{N}_i^\kappa}, a_{\mathcal{N}_j^\kappa}; \boldsymbol{\theta})$ and $\widehat{Q_i^m}(\boldsymbol{s}, a_i; \boldsymbol{\theta})$ in policy gradient approximation. Based on Proposition 1, the approximation error between $\nabla_{\theta_i} J_{app,i}^m(\boldsymbol{\theta})$ and original $\nabla_{\theta_i} J^m(\boldsymbol{\theta})$ in (10) can be well bounded for the MOMARL problem in the following theorem.

**Theorem 1** *In the MOMARL problem, given a joint policy $\boldsymbol{\pi_\theta}$, for any agent $i \in \mathcal{N}$ and objective $m \in \mathcal{M}$, it holds that*

$$\|\nabla_{\theta_i} J_{app,i}^m(\boldsymbol{\theta}) - \nabla_{\theta_i} J^m(\boldsymbol{\theta})\|_2 \leq \frac{\sqrt{2}R}{(1-\gamma^m)^2} (\gamma^m)^{\kappa+1}. \quad (19)$$

Theorem 1 is built upon Lemma 3 and Proposition 1, with its proof provided in Appendix A.4.

The policy gradient has been approximated so far by constructing $\widehat{Q_i^m}(\boldsymbol{s}, a_i; \boldsymbol{\theta})$ in (16) and $\nabla_{\theta_i} J_{app,i}^m(\boldsymbol{\theta})$ in (17), which reduces the action dimension of each agent $i$ to its local action $a_i$. However, the expression of $\widehat{Q_i^m}(\boldsymbol{s}, a_i; \boldsymbol{\theta})$ still requires the global state. Therefore, in the following, we will focus on reducing the dimensionality of agents' state information.

### 3.3 CRITIC STEP: LINEAR FUNCTION APPROXIMATION

As shown in Fig. 1, in this subsection, we use the localized stochastic approximation and propose a linear function in (20) to reduce the dimension of the state-action required by agent $i \in \mathcal{N}$ to $(s_{\mathcal{N}_i^\kappa}, a_i)$. Specially, the linear function $\hat{Q}_i^m(s_{\mathcal{N}_i^\kappa}, a_i; w_i^m)$ of agent $i$ to approximate $\widehat{Q_i^m}(s, a_i; \boldsymbol{\theta})$ is given as

$$\hat{Q}_i^m(s_{\mathcal{N}_i^\kappa}, a_i; w_i^m) = \phi_i(s_{\mathcal{N}_i^\kappa}, a_i)^\top w_i^m, \tag{20}$$

where $\phi_i(s_{\mathcal{N}_i^\kappa}, a_i) : \mathcal{S}_{\mathcal{N}_i^\kappa} \times \mathcal{A}_i \to \mathbb{R}^{d_i}$ is the feature vector mapping and $w_i^m \in \mathbb{R}^{d_i}$ is the parameter of agent $i$ in $m$-th objective. By the definition of $\widehat{Q_i^m}(s, a_i; \boldsymbol{\theta})$ in (16), the parameter with initial value $w_{i,0}^m$ can be updated by sample sequence $\{s_{\mathcal{N}_i^\kappa, t_0}, a_{i,t_0}, r_{\mathcal{N}_i^\kappa, t_0}^m\}_{0 \le t_0 \le K}$ as

$$w_{i,t_0+1}^m = w_{i,t_0}^m - \eta_w^m \delta_{i,t_0}^m \phi_i(s_{\mathcal{N}_i^\kappa, t_0+1}, a_{i,t_0+1}), \tag{21}$$

where $\delta_{i,t_0}^m$ is the local temporal difference error at time $t_0$ and represented as

$$\delta_{i,t_0}^m = \phi_i(s_{\mathcal{N}_i^\kappa, t_0}, a_{i,t_0})^\top w_{i,t_0}^m - \frac{1}{N} \sum_{j \in \mathcal{N}_i^\kappa} r_{j,t_0}^m - \gamma^m \phi_i(s_{\mathcal{N}_i^\kappa, t_0+1}, a_{i,t_0+1})^\top w_{i,t_0}^m, \tag{22}$$

and $\eta_w^m$ is the fixed learning rate of parameters $w_i^m$. The detailed description of linear function approximation is illustrated in Algorithm 2 in Appendix A.5.

### 3.4 ACTOR STEP: POLICY PARAMETER UPDATE

Based on our peoposed approximated policy gradient $\nabla_{\theta_i} J_{app,i}^m(\boldsymbol{\theta})$ in (17), for joint policy $\boldsymbol{\pi}_{\boldsymbol{\theta}_t}$, we denote $g_{i,t}^m(B)$ as the estimation of $\nabla_{\theta_i} J_{app,i}^m(\boldsymbol{\theta})$ based on the sample sequence $\{(s_{\mathcal{N}_i^\kappa, h}^b, a_{i,h}^b)\}_{0 \le b \le B-1, 0 \le h \le H-1}$, calculated by

$$g_{i,t}^m(b+1) = \frac{b}{b+1} g_{i,t}^m(b) + \frac{1}{b+1} \widehat{\nabla}_{\theta_i} J_{app,i}^{m,b}(\boldsymbol{\theta}_t), \tag{23}$$

where $g_{i,t}^m(0) = \mathbf{0}_{|\mathcal{S}_i||\mathcal{A}_i|}$ and $\widehat{\nabla}_{\theta_i} J_{app,i}^{m,b}(\boldsymbol{\theta}_t)$ is defined as

$$\widehat{\nabla}_{\theta_i} J_{app,i}^{m,b}(\boldsymbol{\theta}_t) = \sum_{h=0}^{H-1} (\gamma^m)^h \nabla_{\theta_i} \log \pi_{\theta_{i,t}}(a_{i,h}^b | s_{i,h}^b) \phi_i(s_{\mathcal{N}_i^\kappa, h}^b, a_{i,h}^b)^\top w_{i,t}^m. \tag{24}$$

Let $g_{i,t}^m = g_{i,t}^m(B)^\top$ and $\boldsymbol{g}_t^m = \left( (g_{1,t}^m)^\top, \cdots, (g_{N,t}^m)^\top \right)^\top \in \mathbb{R}^{\sum_{i=1}^N |\mathcal{S}_i||\mathcal{A}_i|}$. Related to Pareto-stationarity in Definition 1, we denote $\widehat{\boldsymbol{\lambda}}_t = (\widehat{\lambda}_t^1, \cdots, \widehat{\lambda}_t^M)^\top \in \mathbb{R}^M$ as solution of the following quadratic programming problem:

$$\min_{\boldsymbol{\lambda}_t = (\lambda_t^1, \cdots, \lambda_t^M)^\top \in \mathbb{R}^M} \left\| \sum_{m=1}^M \lambda_t^m \boldsymbol{g}_t^m \right\|_2^2 \quad \text{s.t. } \boldsymbol{\lambda}_t \ge 0, \|\boldsymbol{\lambda}_t\|_1 = 1. \tag{25}$$

After computing $\widehat{\boldsymbol{\lambda}}_t$, we update the weight $\boldsymbol{\lambda}_t$ as

$$\boldsymbol{\lambda}_t = (1 - \eta_{\boldsymbol{\lambda},t})\boldsymbol{\lambda}_{t-1} + \eta_{\boldsymbol{\lambda},t}\widehat{\boldsymbol{\lambda}}_t, \tag{26}$$

where $\eta_{\boldsymbol{\lambda},t}$ is the learning rate of $\boldsymbol{\lambda}_t$. Denote $\boldsymbol{g}_t = \sum_{m=1}^M \lambda_t^m \boldsymbol{g}_t^m$, the update of $\boldsymbol{\theta}_{t+1}$ is presented as

$$\boldsymbol{\theta}_{t+1} = \boldsymbol{\theta}_t + \eta_{\boldsymbol{\theta},t}\boldsymbol{g}_t, \tag{27}$$

where $\eta_{\boldsymbol{\theta},t}$ is the learning rate of policy parameter. In the NMARL problem, the agents can use $\boldsymbol{\theta}_t$ to achieve the distributed decision based on (1).

## 4 DISTRIBUTED SCALABLE ACTOR-CRITIC ALGORITHM AND ITS PARETO-STATIONARY CONVERGENCE

In this section, we first propose a distributed scalable actor-critic algorithm (i.e., Algorithm 1) for the NMARL problem. Then, we prove the Pareto-stationary convergence of Algorithm 1.

Based on Section 3, we propose a distributed scalable actor-critic algorithm for the MOMARL problem, which is given in Algorithm 1. In order to analyze the Pareto-stationary convergence of Algorithm 1.

---

**Algorithm 1:** Distributed scalable actor-critic algorithm for the MOMARL problem

---

**Require:** The non-negative integers $T$, $B$, $H$, the learning-rates $\eta_w^m$, $\{\eta_{\boldsymbol{\lambda},t}\}_{t\in\{1,\cdots,T\}}$ and $\{\eta_{\boldsymbol{\theta},t}\}_{t\in\{1,\cdots,T\}}$;

**Initialization:** Initialize $\boldsymbol{\lambda}_0 = \frac{1}{M}\mathbf{1}_M \in \mathbb{R}^M$, the policy parameter $\theta_{i,1} \in \mathbb{R}^{|\mathcal{S}_i|\times|\mathcal{A}_i|}$ to follow Gaussian distribution for all $i \in \{1, 2, \cdots, N\}$;

**for** $t = 1, 2, \cdots, T$ **do**

    Initial policy gradient estimation $g_{i,t}^m(0) = \mathbf{0}_{|\mathcal{S}_i||\mathcal{A}_i|}$ for all $i \in \mathcal{N}$;

    **Critic step:** All agents use (21) in Algorithm 2 and output the weight vectors $\{w_{i,t}^m\}_{i\in\mathcal{N}}$;

    **Actor step:**

    **for** $b = 0, 1, 2, \cdots, B - 1$ **do**

        All agents execute the joint policy $\boldsymbol{\pi}_{\boldsymbol{\theta}_t}$ in $H - 1$ horizon;

        Each agent $i \in \mathcal{N}$ collects a sequence of samples, which includes the state information $\{s_j\}_{j\in\mathcal{N}_i^\kappa}$ from its $\kappa$-hop neighbors and its local action information $a_i$, i.e., $\{(s_{\mathcal{N}_i^\kappa,h}^b, a_{i,h}^b)\}_{0\leq h\leq H-1}$;

        Each agent $i$ estimates the local policy gradient in $m$-th objective according to (23);

    **end**

    All agents calculate $g_{i,t}^m = g_{i,t}^m(B)$ by (23) and achieve $\boldsymbol{g}_t^m = \left((g_{1,t}^m)^\top, \cdots, (g_{N,t}^m)^\top\right)^\top$ for all $m \in [M]$;

    Compute $\widehat{\boldsymbol{\lambda}}_t$ as the solution to problem (25);

    Update the weight $\boldsymbol{\lambda}_t$ according to (26);

    Update the policy parameter $\boldsymbol{\theta}_{t+1}$ according to (27);

**end**

**Output:** $\boldsymbol{\pi}_{\boldsymbol{\theta}_{\hat{T}}}$ with $\hat{T}$ chosen uniformly from $\{1, \cdots, T\}$

---

Our process to prove the Pareto-stationary convergence of Algorithm 1 is as follows: (i) We start from the definition of Pareto-stationarity in Definition 2 and analyze the error between the true gradient $\nabla_{\theta_i} J^m(\boldsymbol{\theta}_t)$ and the calculated gradient $g_{i,t}^m$ in (23)(i.e., Lemma 4); (ii) We control $\boldsymbol{\lambda}_t$ by setting the step size $\eta_{\boldsymbol{\theta},t}$ to ensure that Algorithm 1 converges to Pareto-stationary solution in Theorem 2.

**Lemma 4** *In Algorithm 1, for joint policy parameter $\boldsymbol{\theta}_t$, any agent $i \in \mathcal{N}$, and objective $m \in \mathcal{M}$, we have*

$$\mathbb{E}[\|\nabla_{\theta_i} J^m(\boldsymbol{\theta}_t) - g_{i,t}^m\|_2^2] \leq \frac{8R^2}{(1-\gamma^m)^4}(\gamma^m)^{2\kappa+2} + \frac{32}{(1-\gamma^m)^2 B} + \frac{8(\gamma^m)^{2H}}{(1-\gamma^m)^4} + \frac{8\varepsilon_{critic}^{\boldsymbol{\theta}_t}}{(1-\gamma^m)^2},$$

*where $\varepsilon_{critic}^{\boldsymbol{\theta}_t}$ is the linear approximation error and defined as*

$$\varepsilon_{critic}^{\boldsymbol{\theta}_t} = \sup_{m\in\mathcal{M}} \sup_{i\in\mathcal{N}} \mathbb{E}\left[\sup_{\boldsymbol{s},a_i}\left|\hat{Q}_i(s_{\mathcal{N}_i^\kappa}, a_i; w_{i,K}^m) - \widehat{Q_i^m}(\boldsymbol{s}, a_i; \boldsymbol{\theta}_t)\right|^2\right]. \tag{28}$$

The proof of the Lemma 4 is given in Appendix A.6. Based on Lemma 4, the Pareto-stationary convergence of Algorithm 1 is presented in the following theorem.

**Theorem 2** *In Algorithm 1, let $L_J = \max_{m\in\mathcal{M}} \frac{6N}{(1-\gamma^m)^3}$, $\eta_{\boldsymbol{\theta},t} = \frac{1}{3L_J}$, and $\eta_{\boldsymbol{\lambda},t} = \frac{1}{(t+1)^2}$. Our policy parameter sequences $\{\boldsymbol{\theta}_t\}_{t=1}^T$ generated by Algorithm 1 satisfies:*

$$\mathbb{E}[\|\nabla_{\boldsymbol{\theta}}\boldsymbol{J}(\boldsymbol{\theta}_{\hat{T}})^\top\widehat{\boldsymbol{\lambda}}_{\hat{T}}\|_2^2] \leq \frac{36L_J}{(1-\|\gamma\|_\infty)T}\left(1 + \sum_{t=1}^T \eta_{\boldsymbol{\lambda},t}\right) + 5\max_{m\in\mathcal{M}}\left(\frac{8R^2}{(1-\gamma^m)^4}(\gamma^m)^{2\kappa+2}\right.$$

$$\left. + \frac{32N}{(1-\gamma^m)^2 B} + \frac{8(\gamma^m)^{2H}N}{(1-\gamma^m)^4} + \frac{8\max_{1\leq t\leq T}\varepsilon_{critic}^{\boldsymbol{\theta}_t}N}{(1-\gamma^m)^2}\right), \tag{29}$$

*where $\hat{T}$ is uniformly sampled among $\{1, \cdots, T\}$.*

The proof of Theorem 2 can be found in Appendix A.7. Theorem 2 shows that Algorithm 1 can converge to an approximate Pareto-stationary solution at a rate of $\mathcal{O}(1/T)$. The gap between the approximate Pareto-stationary and the Pareto-optimal depends on graph-truncated approximation error $\frac{8R^2}{(1-\gamma^m)^4}(\gamma^m)^{2\kappa+2}$ and linear function approximation error $\frac{8\varepsilon_{critic}^{\theta_t}N}{(1-\gamma^m)^2}$. These errors are not significant, as we can control the upper bound of their upper bounds by setting the graph-truncated distance $\kappa$ and the feature vector in the linear approximation. Specially, the graph-truncated approximation error is exhibits an exponential decrease as $\kappa$ increases.

## 5 ROBOTS PATH PLANNING EXPERIMENTS

In this section, we study MOMARL by considering $N$ robots as agents in a typical path planning simulation experiment by following (Zhou et al., 2023). Similar setting is also used in (Duan et al., 2016; Zhang & Pavone, 2016). We consider different path networks as shown in Figs. 2(a) and 3(a), where leftmost nodes represent the starting locations for agents and rightmost nodes represent the different objective destinations. The agents have the option to either halt or continue along the path until they reach the objective destinations, where they will remain. The goal of agents is to explore different destinations, for simultaneously minimizing the travel time and collision with each other.

In path planning simulation experiment, for each agent $i \in \{1, \cdots, N\}$, define all possible locations as its local state space and all possible movements as its local action space. In order to better understand the movement changes of agents, we take network 3-2-2 in Fig. 2(a) as an example. If agent $i$ at node $b_2$, it can choose remain stationary at the current node for one time step, move along the edge $(b_2, c_1)$ or edge $(b_2, c_2)$.

The reward setting of each agent $i$ includes: (i) the cost of travel time $-0.5$ at each step, (ii) the collision penalty $-0.5$ when it chooses the same path with another to move, (iii) the final reward for reaching a destination. Specifically, when a agent reaches objective 1 and objective 2 in network 3-2-2, it will receive additional rewards of $[0.5, 0]$, and $[0, 1]$, respectively. In network 5-5-5-3, each agent reaches objective 1, objective 2, and objective 3 will receive the additional rewards of $[0.5, 0, 0]$, $[0, 1.5, 0]$, and $[0, 0, 1]$, respectively. The goal of agents is to find a joint policy parameter $\boldsymbol{\theta}$ to maximize (3).

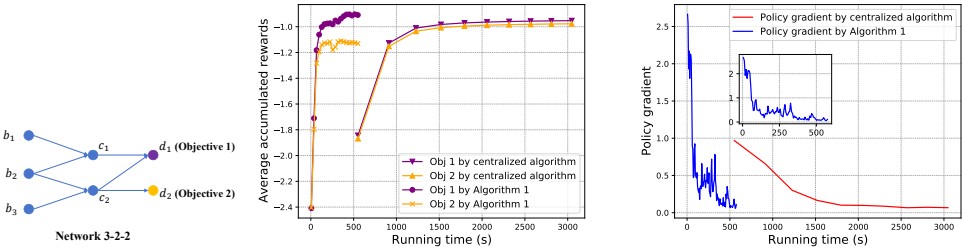

(a) Acyclic network  (b) The evolution of the objective performance $J(\boldsymbol{\theta}_t)$  (c) The evolution of the norm of policy gradient $\|\boldsymbol{g}_t\|_2$

Figure 2: (a) Experiment network setting for $N = 6$ robots, (b) the multi-objective performances, and (c) the norm of gradient of our Algorithm 1 as compared to the centralized Algorithm 3.

In path network 3-2-2, we set the discount factor $\boldsymbol{\gamma} = (0.9, 0.9)^\top$, the communication distance $\kappa = 1$, and the initial positions of agents are set to $b_1, b_2, b_3, b_1, b_2, b_3$, respectively. In order to demonstrate the superiority of our proposed Algorithm 1 in terms of runtime and computational performance, we compare it to the centralized Algorithm 3 presented in Appendix A.8, which uses the global state-action information and has also been proven to converge to 0-Pareto-stationarity (i.e., Theorem 4 in Appendix A.8).

The discounted average cumulative reward $\{J^m(\boldsymbol{\theta}_t)\}_{m \in \{1,2\}}$ of the policy sequence generated by Algorithm 1 and the centralized Algorithm 3 are depicted in Fig. 2(b), where x-axis represents the running time. Although the final value of objective 2 generated by centralized Algorithm 3 is better than Algorithm 1, it takes longer time to learn. As shwn in Fig. 2(b), centralized Algorithm 3 takes 575s to implement an update to the policy parameters, but our algorithm has already learned in this time. Furthermore, the value of objective 1 in our proposed Algorithm 1 converges to greater value as compared to the centralized Algorithm 3.

The Pareto-stationary convergence error (i.e., $\|\boldsymbol{g}_t\|_2$ in (27)) generated by Algorithm 1 and the centralized Algorithm 3 is depicted in Fig. 2(c), where the x-axis represents the running time. Although the norm of policy gradient generated by centralized Algorithm 3 is closer to 0 than Algorithm 1, the norm of policy gradient of our Algorithm 1 can reach to 0.05 quickly after running 575s, which is significantly faster than the centralized Algorithm 3. This speed advantage stems from the fact that the centralized algorithm requires time-consuming calculations of the exact value of the global $Q$-function during policy updates. In contrast, our Algorithm 1 does not necessitate such computations and thus outperforms the centralized algorithm in term of runtime.

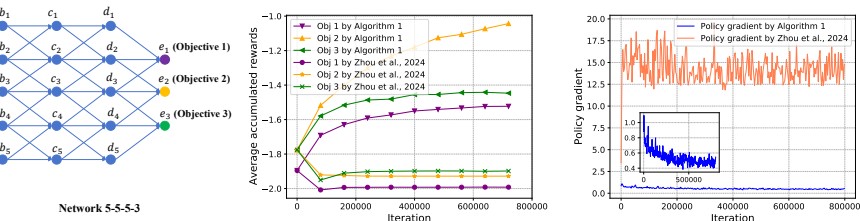

|(a) Acyclic network | (b) The evolution of the objective performance $J(\boldsymbol{\theta}_t)$ | (c) The evolution of the norm of policy gradient $\|\boldsymbol{g}_t\|_2$|

Figure 3: (a) Experiment network setting for $N = 10$ robots, (b) the multi-objective results, and (c) the norm of gradient of our Algorithm 1 as compared to the latest MORL algorithm (Zhou et al., 2024).

In the larger path network 5-5-5-3, we set the discount factor $\boldsymbol{\gamma} = (0.9, 0.9, 0.9)^{\top}$, the communication distance $\kappa = 1$, and the initial positions of agents are set to $b_1, b_2, b_3, b_4, b_5, b_1, b_2, b_3, b_4, b_5$, respectively. In this simulation, the centralized Algorithm 3 is no longer applicable due to its enormous computational complexity. Thus, we compare our Algorithm 1 to the latest MORL algorithm (Zhou et al., 2024), which specifically addresses the MORL problem with discrete action space and is currently the only approach for achieving Pareto-stationarity. Since the latest MORL algorithm cannot directly apply to our multi-agent setting of limited communications, we transform the multi-agent setting to its MORL with a single agent, who accesses the global state-action information.

The discounted average cumulative reward $\{J^m(\boldsymbol{\theta}_t)\}_{m \in \{1,2,3\}}$ of the policy sequence generated by our Algorithm 1 and the latest MORL algorithm are depicted in Fig. 3(b), where x-axis represents the number of iterations. As shown in Fig. 3(b), our Algorithm 1 converges to all greater multi-objective values as compared to the latest MORL algorithm.

In order to demonstrate the superiority of the algorithm in convergence performance, the Pareto-stationary convergence error generated by Algorithm 1 and the latest MORL algorithm are shown in Fig. 3(c), where the x-axis represents the number of iterations. The norm of the policy gradient, as demonstrated by Algorithm 1, exhibits a clear convergence trend towards 0. However, the policy gradient in the latest MORL algorithm deviates significantly from 0 due to the excessively large global state-action dimension, resulting in a substantial approximation error in the global $Q$-function approximation.

Based on the simulation results, the centralized Algorithm 3 necessitates the computation of the exact value of the global $Q$-function at each update, resulting in a time-consuming procedure. The latest MORL algorithm (Zhou et al., 2024) employs an approximation of the global $Q$-function, which enhances its efficiency; however, it encounters convergence challenges in MAMORL problem. In comparison to the centralized Algorithm 3 and the latest MORL algorithm (Zhou et al., 2024), our proposed Algorithm 1 demonstrates favorable outcomes in terms of both running time and convergence.

## 6 CONCLUSIONS

In this paper, we proposed a distributed scalable actor-critic algorithm for the MOMARL problem and proved that this algorithm reaches a close-to-Pareto-stationary point of $\boldsymbol{J}(\boldsymbol{\theta})$. In the proposed algorithm, each agent only requires state-action information $(s_{\mathcal{N}_i^\kappa}, a_i)$, which can effectively improve the scalability of the algorithm. The underlying framework of distributed scalable actor-critic algorithm, which includes the graph-truncated $Q$-function (12) and the action-averaged $Q$-function (16), constitutes a significant contribution in its own right and has the potential to pave the way for other scalable reinforcement learning methods in networked systems.

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
