# OpenReview forum: "Multi-objective Multi-agent Reinforcement Learning with Pareto-stationary Convergence"
_ICLR.cc/2025/Conference — ICLR 2025 Conference Withdrawn Submission_

### Official Review · Reviewer_YgQE · 2024-10-31

**Soundness:** 3
**Presentation:** 3
**Contribution:** 2
**Rating:** 6
**Confidence:** 2

**Summary:**

This paper proposes a new solution concept to the challenging class of multi-objective multi-agent problems. The authors combine graph-truncated Q-functions and action-averaged Q-functions with a linear function approximation to obtain their learning algorithm. The derivations are based on theoretical results. The paper concludes with an illustrative robot path planning problem to demonstrate the capabilities of the proposed approach.

**Strengths:**

-	The introduction is well-written. It provides intuitive examples, gives an overview of the existing literature and states open problems. Finally, the authors clearly pose their research question and their contributions.
-	The paper is structured convincingly. Although the mathematical notations are quite extensive, readers can follow the basic ideas because the authors frequently explain their next steps and theoretical findings.
-	The algorithmic findings are complemented by extensive theoretical results. Furthermore, the proposed approach seems to outperform an existing method on the given robot path planning example.

**Weaknesses:**

-	The paper focuses on finding Pareto-stationary solutions, as the authors state in line 161, instead of Pareto-optimal solutions. While I understand the authors’ reasoning and their arguments for focusing on stationarity, it still is a rather severe limitation. Is there any possibility to extend the method to Pareto optimality (other than assuming a convex problem)?
-	Lemma 2 appears to be almost identical to Lemma 3 in (Qu et al., 2020a) and therefore provides no new significant insights. Also, the appendix section title “A.1 The Detailed Proof of Lemma 2” suggests a detailed proof but the proof essentially only refers to the existing result of (Qu et al., 2020a) which makes the section title misleading in my opinion.
-	As the authors themselves state, Lemma 3 is also similar to results in (Qu et al., 2020a). This raises the question whether there are any substantial novelties in Subsection 3.1. The authors should explain more precisely how their results relate to those in (Qu et al., 2020a).
-	The experiment section is somewhat limited because there is just one robot example on two rather small networks with few agents. It would be helpful if the authors could include further examples and elaborate on the scalability of their algorithm with respect to the number of agents and the size of the network.

Minor comments:

-	Maybe remove the mathematical expressions from the abstract, e.g., just write “state-action” and skip “(s, a)” in the abstract
-	Line 74: introduce the neighborhood state-action notation $(s_{\mathcal N}, a_{\mathcal N})$ before using the mathematical expression. I would suggest to just move the mathematical terms like $(s_{\mathcal N}, a_{\mathcal N})$ to Section 2.
-	Page 2, footnote: Personally, I would remove the footnote and include the information either in the main text or defer it to the appendix.
-	Line 98 and following: Are there any restrictions or assumptions on the local state and action space. For example, are they finite or continuous?
-	Line 116: If I understand correctly, $s_0$ refers to the initial state of the whole system and not just the state of one agent. Could you emphasize this by adding something like $s_0 \in \mathbb{S}$?
-	Line 131: Maybe I missed it, but are $S_i$ and $A_i$ assumed to be finite?
-	Figure 1: The font size is very small which makes it hard to read the figure. This is especially unfortunate since the figure seems to visualize many of the key ideas in the paper and provides an important overview.
-	In general, the authors could mention the assumptions underlying each theoretical result, for example: “Lemma 2: Under Assumption 2, the MOMARL problem satisfies …”
-	To my understanding, Theorem 1 is immediately obtained by just plugging equality (18) into inequality (15). I am not sure if this requires a detailed proof in Appendix A.4.
-	There are some typos in the paper, such as “peoposed” (line 247) and “shwn” (line 483)
-	Lines 378-379: Isn’t the sentence “In order to analyze the Pareto-stationary convergence of Algorithm 1.” missing a second sentence part?

**Questions:**

-	Since I am not an expert in the MOMARL literature, I am wondering whether there exist other approaches to MOMARL problems (apart from the one mentioned in lines 62 to 66)?
-	How crucial is the assumption of softmax policies for the paper? How would other policies work?
-	Definition 2, Lemma 1: Are there any assumptions on $J$ or $r$ to ensure that the gradient exists?
-	Why is the MORL algorithm of (Zhou et al., 2024) just used for the larger network? Wouldn’t it make sense to also include it as a third option in the first network?
-	The MORL algorithm of (Zhou et al., 2024) seems to perform worse than the initialization in Figure 3 (b) and therefore appears to learn nothing useful. Is this behavior justifiable by the “approximation of the global Q-function” (line 526)?
-	How well does the proposed approach scale with the number of agents $N$ and the size of the network?

---

### Official Review · Reviewer_cH7H · 2024-11-01

**Soundness:** 3
**Presentation:** 2
**Contribution:** 3
**Rating:** 5
**Confidence:** 2

**Summary:**

This paper focuses on the problem of multi-agent multi-objective optimization through multi-agent reinforcement learning. In specific, the authors seek to find a scalable methodology to find a set of multi-agent Pareto-stationary solutions (e.g. solutions where no objective can be unilaterally improved without sacrificing another). The authors propose a graph truncated Q-function approximator and action-averaged Q-function for policy gradient approximation. They use a linear approximator for the action averaged Q-function, thereby reducing the dimensionality of the state. Through proofs and experiments, they demonstrate the convergence properties of their algorithm and improved computational efficiency.

**Strengths:**

The authors were thorough with their proofs for each algorithm component. The timing tests across algorithms effectively demonstrated the algorithm's computational efficiency. The contribution of this work is clearly stated.

**Weaknesses:**

In the robot path planning experiment section, the experimental section could be strengthened with additional comparisons against other algorithms in the literature. For example, MO-MIX [1], MOMAPPO [2], and PACDCG [3] could be interesting points to compare. Similarly, there could have been more references to existing MOMARL work in the introduction beyond existing work in single agent MORL.  It also would be helpful to add further context about the environment simulator and the associated objective function parameters selected, and reasoning behind the graph structure.

[1] MO-MIX: Multi-Objective Multi-Agent Cooperative Decision-Making with Deep Reinforcement Learning, Hu et al. 2023
[2] MOMALand: A Set of Benchmarks for Multi-Objective Multi-Agent Reinforcement Learning, Felten et al. 2024.
[3] Pareto Actor-Critic for Equilibrium Selection in Multi-Agent Reinforcement Learning, Christianos et al. 2024

**Questions:**

- Can you explain more about the reward formulation mentioned in Section 5? I understand that the vector size for the reward is shaped by the number of objectives, but how is the reward value itself selected?
- In Figure 3, how many runs were used to generate the reward curve (b) and policy gradient curve (c)? Is it possible to show the standard deviation across runs?
- Can you elaborate more on the simulation experiment you used? The Zhou et. al 2023 paper mentions the simulation they use in the appendix, but not a detailed description of the simulator itself.
- Why did you select 3-3-2 vs 5-5-5-3 graph structures? How does 5-5-5-3 challenge the algorithm beyond increase in computational complexity?
- How does this method extend to sparse graphs with lower K values? Are there any assumptions on the level of connectivity for the graph network?

---

### Official Review · Reviewer_29XQ · 2024-11-05

**Soundness:** 2
**Presentation:** 3
**Contribution:** 2
**Rating:** 3
**Confidence:** 4

**Summary:**

The paper presents a linear actor-critic-based algorithm for multi-objective multi-agent reinforcement learning. The authors claim to achieve a $1/T$-Pareto stationarity by allowing the agents to collect their neighbors' information in $T$ iterations.

**Strengths:**

1. Good presentation and write-up.

**Weaknesses:**

**Minor Comments**

1. The notations $\mathcal{S}\_{-\mathcal{N}}$ and $\mathcal{A}\_{-\mathcal{N}}$ are not properly defined/introduced anywhere.

2. The concept of exponential decay property is quite old (at least in single-objective MARL setups). The authors should give proper citations to it.

**Major Comments**

3. Although the authors claim that the agents need to communicate only with their neighbors, I see that some parts of the algorithms are centralized, e.g., the updates of the policy parameter $\boldsymbol{\theta}$ and the Pareto parameter $\boldsymbol{\lambda}$. This should be highlighted in the introduction/contribution part of the paper.

4. How is the optimization $(25)$ solved that computes $\hat{\boldsymbol{\lambda}}$? Is it solved by a single agent in a centralized manner or is it done in a decentralized fashion? This should be clarified in the paper. Since nothing is mentioned, I will assume this has to be done in a centralized fashion.

5. If $(25)$ is indeed solved in a centralized fashion, does it not violate the main motivation of the paper i.e., the agents only need to know their neighbours' information and not everyone else's?

6. Theorem 2 shows that the gradient error bound is $\mathcal{O}\left(\frac{1}{T}+\frac{N}{B}+\gamma^{2H}\right)$ where $T$, $B$, $H$ are explained in the paper. However, both in the introduction and in the abstract, the authors present the error as $\mathcal{O}\left(\frac{1}{T}\right)$ and ignore other factors. Why so? In a sample-based learning setup, the sample complexity (determined by $T, B, H$) is more important than the iteration complexity (determined solely by $T$).

7. A proper communication complexity analysis is missing.

**Questions:**

1. Is Assumption 1 connected to the ergodicity assumption typically applied in the policy gradient-type analysis?

2. Where is the critic approximation error formally defined in the main text of the paper?

---

### Official Review · Reviewer_ESEE · 2024-11-08

**Soundness:** 2
**Presentation:** 1
**Contribution:** 3
**Rating:** 5
**Confidence:** 3

**Summary:**

The paper presents a novel algorithm for multi-objective multi-agent reinforcement learning (MOMARL) via graph-truncated Q-function approximation method, which only requires local state-action information from each agent's neighborhood rather than global data. Additionally, they introduce the concept of an action-averaged Q-function, reducing the dimensionality further to local states and actions, and establish an equivalence between the graph-truncated Q-function and action-averaged Q-function for policy gradient approximation. They develop a distributed, scalable algorithm with linear function approximation and prove it converges to a Pareto-stationary solution at a rate of O(1/T).

**Strengths:**

1. By using a graph-truncated Q-function that only relies on local state-action information, the algorithm avoids the exponential growth of the global state-action space.
2. The algorithm is mathematically proven to converge to a Pareto-stationary solution at a rate of O(1/T).

**Weaknesses:**

1. Writing is rush and poor.
2. The main conclusions and theoretical results are introduced relatively late, with the first major lemma only appearing on page 4 in a 10-page paper. This delay may reduce the immediate impact and engagement for readers, as foundational results that frame the work are postponed.
3. Many equations are labeled even though they are referenced only once. This creates unnecessary clutter and can impede readability. Reducing labels to only frequently referenced equations would improve flow and make the reading experience smoother.
4. The expression of the goal in multi-agent, multi-objective reinforcement learning (MPMARL) is unclear, particularly in maximizing a vector with potentially correlated values.
5. Definition 2 (ε-Pareto-stationarity) lacks citation of relevant articles. Providing references to foundational works on Pareto-stationarity, along with an explanation, would help readers connect the definition to established literature.
6. Assumptions 1 and 2 are verbose, which affects conciseness and clarity.
7. Lemma 1 lacks a proof or reference to an appendix section, as well as citations of relevant works. This absence is also in Lemma 3.
8. The start of Section 3 does not provide sufficient motivation for addressing the algorithm’s reliance on the global state-action space.

**Questions:**

See in weakness.

---

### Note · Authors · 2024-11-28

**Comment:**

Withdraw this manuscript.

**Withdrawal Confirmation:**

I have read and agree with the venue's withdrawal policy on behalf of myself and my co-authors.